# *Dyrk1a* from Gene Function in Development and Physiology to Dosage Correction across Life Span in Down Syndrome

**DOI:** 10.3390/genes12111833

**Published:** 2021-11-20

**Authors:** Helin Atas-Ozcan, Véronique Brault, Arnaud Duchon, Yann Herault

**Affiliations:** 1Université de Strasbourg, CNRS, INSERM, Institut de Génétique et de Biologie Moléculaire et Cellulaire (IGBMC), 1 rue Laurent Fries, 67404 Illkirch Graffenstaden, France; atash@igbmc.fr (H.A.-O.); vbrault@igbmc.fr (V.B.); duchon@igbmc.fr (A.D.); 2Université de Strasbourg, CNRS, INSERM, Celphedia, Phenomin-Institut Clinique de la Souris (ICS), 1 rue Laurent Fries, 67404 Illkirch Graffenstaden, France

**Keywords:** trisomy 21, neurodevelopmental disorder, mouse model, cognition, learning and memory, preclinical trial

## Abstract

Down syndrome is the main cause of intellectual disabilities with a large set of comorbidities from developmental origins but also that appeared across life span. Investigation of the genetic overdosage found in Down syndrome, due to the trisomy of human chromosome 21, has pointed to one main driver gene, the *Dual-specificity tyrosine-regulated kinase 1A (Dyrk1a).* Dyrk1a is a murine homolog of the drosophila *minibrain* gene. It has been found to be involved in many biological processes during development and in adulthood. Further analysis showed its haploinsufficiency in mental retardation disease 7 and its involvement in Alzheimer’s disease. DYRK1A plays a role in major developmental steps of brain development, controlling the proliferation of neural progenitors, the migration of neurons, their dendritogenesis and the function of the synapse. Several strategies targeting the overdosage of DYRK1A in DS with specific kinase inhibitors have showed promising evidence that DS cognitive conditions can be alleviated. Nevertheless, providing conditions for proper temporal treatment and to tackle the neurodevelopmental and the neurodegenerative aspects of DS across life span is still an open question.

## 1. Introduction

*DYRK1A* encodes the dual-specificity tyrosine-regulated kinase 1A whose role in physiology and disease has been pinpointed over the years. This gene located on Human chromosome 21 (Hsa21) has benefited from the pioneering study of the *minibrain* mutant, highlighting the role of *DYRK1A* drosophila homologs in neuronal development. This major role has fostered research on the consequence of its overdosage in Down syndrome (DS), the major causes of intellectual disability. Further investigation has led to the discovery of the *DYRK1A* haploinsufficiency syndrome (also named Mental Retardation autosomal Dominant 7, MRD7). *DYRK1A* is deeply regulated both at the expression and kinase level while expressed in almost every cell. In addition, DYRK1A has been involved in many different cellular and molecular processes, from cell proliferation to synaptic function, from gene expression regulation to the control of pathways for apoptosis and neurodegeneration. *DYRK1A* is definitely a main gene driver of cognitive impairment in DS models, and DYRK1A inhibitors have emerged as a promising therapeutic path to reduce DS cognitive deficits in adults but also after in utero treatment in animal models. In addition recent advances in kinase inhibitors are ongoing with the definition of proper temporal window treatment to tackle the neurodevelopmental and the neurodegenerative aspect of DS over life span.

## 2. DYRK1A a Kinase with Multiple Roles at the Frontiers of Neuronal Proliferation, Differentiation, and Function

### 2.1. DYRK1A, a Unique Member of a Kinase Subfamily

#### 2.1.1. The Family of DYRK Proteins

DYRK1A (dual-specificity tyrosine-regulated kinase 1A) is a member of the dual-specificity tyrosine-regulated kinases (DYRKs) family that belongs to the GMGC kinase group, including cyclin-dependent kinases (CDK), mitogen-activated protein kinases (MAPK), glycogen synthase kinase (GSK3), and the CDC-like kinase (CLK) group of protein kinases. These GMGC kinases are having pivotal roles in cell cycle regulation, intracellular signal transduction, cell growth and communication [1]. A unique feature of the DYRKs kinase domain separates them from two other close subfamilies: the homeodomain-interacting protein kinases (HIPKs) and the protein kinases 4 (PRP4) involved in regulation of pre-mRNA splicing. Since the discovery of the Yak1, the founder of the DYRK family in yeast [2], members of the DYRK subfamily have been found in all eukaryotes and are divided into two classes originating from a genetic duplication event that has occurred during metazoan evolution. DYRK1A belongs to class I DYRKs that includes also mammalian DYRK1B, *Caenorhabditis* MBK1 and drosophila minibrain (Mnb) [3], while class II includes mammalian DYRK2, DYRK3 and DYRK4, *C. elegans* MBK2 and drosophila dDyrk2 and dDyrk3. In addition to its kinase domain, DYRK1A shares with other members of the mammalian DYRKs a conserved N-terminal motif (DH-box for Dyrk Homology box) which might provide a chaperon-like function during protein maturation that stabilizes the kinase domain in an intermediate form with tyrosine autophosphorylation activity [4]. DYRK1A additionally shares with DYRK1B a nuclear localisation signal (NLS) recognized in the cytosol by the importin protein, which guides proteins towards the nuclear pores, and a PEST protein degradation signal sequence, a hallmark of short-lived proteins. Finally, a polyhistidine tract that targets the protein to splicing factor compartments within the nucleus is present in the specific C-terminal part of DYRK1A [5]. The crystal structure of the kinase domain has been obtained by co-crystallization with different inhibitors such as harmine or INDY (inhibitor of DYRK1A), showing that the N-terminal and C-terminal lobes of the catalytic domain are connected by a hinge region forming the ATP-binding pocket and containing phenylalanine 238 as a gatekeeper responsible for the selectivity of the inhibitors [6,7,8,9,10,11,12].

#### 2.1.2. DYRK1A Expression and Expression Regulation

DYRK1A is encoded on the Human chromosome 21 (HSA21) in 21q22.2 and codes at least five isoforms, with two main proteins of 763 and 754 amino acids (aa), the 763 aa being ubiquitously expressed in tissues during development and in the adult, with a stronger expression in the central nervous system (CNS) [13,14], the other isoforms being rare in the brain [15,16,17]. During CNS development, *Dyrk1a* is first transiently detected in pre-neurogenic proliferative cells in 8 to 10.5 days postcoitum (dpc) mouse embryos. At 15.5 dpc, DYRK1A is present in cycling neuronal progenitor cells in the ventricular and subventricular zones and in differentiated neurons [18]. DYRK1A expression peaks during neuronal dendritic morphogenesis at birth [18,19] and remains expressed at lower levels during adulthood [13]. At the adult stage, DYRK1A is present throughout the brain with higher expression in the olfactory bulb, the piriform/entorhinal cortices, cerebellar cortex, periaqueductal gray and pontine nuclei, the cerebellar cortex and neurons of the oculomotor and trigeminal motor nuclei [20]. DYRK1A is not only present in neurons, but also to a lesser extent in astrocytes and microglia [20,21].

Two non-overlapping promoters that start the transcription of two splicing variants of human *DYRK1A* have been described by Maenz et al. [22], although no evidence of cell- or tissue-specific use of those promoters was found. They, however, showed a specific regulation of one of the two promoters by the E2F1 transcription factor known to regulate the expression of genes involved in cell cycle progression [22]. The REST silencing transcription factor whose expression is coordinated with that of DYRK1A during neuronal development was shown to activate *DYRK1A* via a neuron-restrictive silencer element (NRSE) present at 833 to 815 bp of the human *DYRK1A* promoter, forming a negative feedback loop with DYRK1A, that can reduce its transcriptional activity [23]. Additional transcription factors can regulate *DYRK1A.* First, the transcription factor OLIG2, with a known homolog on the Hsa21, was shown by ChIP-Seq analysis to target enhancer regions of *Dyrk1a*, inhibiting its expression [24]. *Olig2* overdosage also contributes to Down syndrome developmental brain defects in mouse models [25], and its overexpression in neural progenitors causes a precocious cell cycle exit and neuronal cell death. Then, *DYRK1A* mRNA translation is MENA-dependent in developing axons. MENA is an actin regulatory protein occurring both at steady state and after BDNF stimulation and emphasizing a requirement of DYRK1A in axon guidance and synaptogenesis [26]. P53, the tumor suppressor protein, and P44, a shortest form, whose overexpression causes premature aging in mice [27] with cognitive decline, synaptic defects, and MAPT/TAU hyperphosphorylation [28], were found to bind to the promoter of *Dyrk1a* and activate its transcription as well as the transcription of other kinases known to be involved in MAPT/TAU phosphorylation [29]. DYRK1A can also phosphorylate p53, leading to the induction of p53 target genes and affecting the proliferation of rat neuronal cell lines [30].

*Dyrk1a* expression is also regulated by several miRNAs. The *Dyrk1a* mRNA level is reduced by miR-1246 whose expression is induced by p53 in response to DNA damage [31]. Overexpression of P53 in human lung carcinoma cells induced *miR-1246* expression which targeted DYRK1A mRNA 3′UTR, reducing DYRK1A protein levels. As the consequence of DYRK1A decreases, NFATC1, a target of DYRK1A, was no longer exported out of the nucleus leading to dramatic cell apoptosis in cancer cells [31]. miR-199b, another miR under the control of NFATc, also targets DYRK1A mRNA, leading to hypertrophy and fibrosis in a mouse model of heart failure [32]. Finally, a recent report identified miR-204-5p as a marker overexpressed in the serum of patients with Parkinson’s disease and proposed from data analyses that overexpression of this miR positively regulates DYRK1A mRNA expression, leading to the death of dopaminergic cells via DYRK1A-mediated ER stress and apoptotic signaling cascade [33].

#### 2.1.3. Regulation of DYRK1A Kinase Activity

DYRKs kinases have the particularity to be dual-specificity protein kinases, phosphorylating on tyrosine, serine, and threonine residues. In those proteins, unlike mitogen-activated protein kinases (MAPK), the kinase activity does not require an upstream activating kinase, but the kinase is constitutively activated by an autocatalytic process. DYRK1A autophosphorylates its second tyrosine residue (Tyr321) of its activation loop YxY motif during translation while bound to the ribosome, without the presence of chaperone proteins [34]. This tyrosine autophosphorylation produces an active DYRK1A with enhanced serine/threonine kinase activity [35], which cannot be inactivated by dephosphorylation of Tyr321 [36]. Based on studies of the drosophila DYRKs (MNB and dDYRK2), DYRK1A was first thought to lose its tyrosine phosphorylation ability after maturation [37]. However, Walte and collaborators showed that mature DYRK1A was still capable of tyrosine phosphorylation and that this activity is not limited to activating-autophosphorylation as they identified Tyr104 and Tyr111 in the N-terminal region as phosphorylated sites [38]. Additional autophosphorylation sites (Tyr140 and Tyr159 in the DH-box region, Tyr177 in the N-terminal lobe, and Tyr449 near the C-terminus) were found [11]. Moreover, an autophosphorylation of Ser97 was found to occur during the translation or folding process, before the occurrence of the intermediated folding form of DYRK1A autophosphorylating Tyr321. This phosphorylation stabilizes DYRK1A by blocking its degradation without affecting its catalytic activity [39]. In an in vitro analysis, autophosphorylation of Ser520 was found to be required for the interaction of DYRK1A with the 14-3-3 kinase-binding protein, this interaction leading to an increase of the catalytic activity of DYRK1A [40]. Similarly, autophosphorylation of Ser529 within the C-terminal PEST domain was shown to trigger a conformation change of DYRK1A that enabled it to interact with the 14-3-3β protein, although this interaction was not found to have any effect on intrinsic DYRK1A kinase activity [41]. Tyr 145 and 147 phosphorylated and nonphosphorylated forms of DYRK1A were described in the brains of human and mouse with DS, with phosphorylation of those two tyrosines being abundant in the neuronal cytoplasm of postnatal brains but low in nuclei of astroglial cells, adult hippocampal progenitors and some cholinergic axon terminal [42]. The nonphosphorylated Tyr-145/147 form was found to be more abundant in the trisomic brain, although the functional consequence of this phosphorylation was not elucidated [42]. Moreover, some phosphotyrosines (Tyr104, Tyr145 and Tyr147) have been described in the N-terminal domain of DYRK1A outside its activation domain by phosphoproteomics analyses (www.phosphosite.org, accessed on 1 November 2020).

SPRED1 and SPRED2, two members of the Sprouty family, that are phosphorylation substrates for tyrosine kinases in response to growth factors, were found to decrease DYRK1A phosphorylation activity on TAU and STAT3. They bind to the DYRK1A kinase domain and compete for the phosphorylation binding site for DYRK1A substrates [43]. Although DYRK1A is believed to be constitutively active after its autophosphorylation on Tyr321, phosphorylation and dephosphorylation of DYRK1A by other kinases and phosphatases have been observed. Yang et al. [44] found that bFGF was selectively activating DYRK1A by tyrosine phosphorylation, leading to gene transcription via activation of the CREB transcription factor during differentiation of hippocampal progenitor cells. DYRK1A was found to contain putative consensus phosphorylation sites for the LATS2 kinase (HxH/R/KssS/T), a component of the hippo pathway that plays a pivotal role in organ size control and tumor suppression by restricting proliferation and promoting apoptosis. Moreover, Tschop and collaborators [45] showed that LATS2 phosphorylates DYRK1A in a dose dependent manner, leading to the phosphorylation by DYRK1A of the LIN52 subunit of the DREAM complex and promoting the assembly of this repressor complex at E2F-regulated promoters. Most recently, the protein phosphatase PPM1B was found to dephosphorylate the DYRK1A Ser258 auto-phosphorylated residue, inhibiting the catalytic activity of DYRK1A and its phosphorylation of TAU, reducing toxic TAU aggregation [46]. Enhanced kinase activity can also be obtained by DYRK1A proteolysis. The calcium-activated cysteine protease calpain I has been found to proteolyze DYRK1A at its C-terminus, probably within its PEST region, resulting in a truncated form found to be increased in the brain of Alzheimer’s patients and enhancing DYRK1A kinase activity toward TAU [47]. Furthermore, Souchet and collaborators [48] demonstrated that the cleaved form of DYRK1A is present in astrocytes in the hippocampus of AD patients and APP/PS1 mice, although proteolysis in their case was not associated with a modification of global DYRK1A kinase activity but was associated with a reduced kinase specificity and a stronger affinity towards STAT3a, an activator of neuroinflammation. The presence of a PEST sequence in DYRK1A also suggests a rapid turnover of the protein that can also contribute to the regulation of DYRK1A activity. In HEK293 (human embryonic kidney 293) cells, DYRK1A was observed to be degraded through the ubiquitin proteasome pathway during cell cycle progression, although this was not achieved through the PEST region but through a N-terminus unconserved binding motif (^49^SDQQVSALS^57^) [49].

Another key mechanism for regulation of DYRK1A activity is the control of its subcellular localization that enables to control the accessibility to its different substrates. DYRK1A was first classified as a nuclear protein [50], consistent with the two NLS signals found in the protein. However, analysis of biochemical fractions of mouse brains revealed that DYRK1A is present in both the nucleus and cytoplasm in neuronal cells, while it was only found to be mostly present in the cytoplasm of glial cells [19,20,21,51]. Overexpression of DYRK1A resulted in the accumulation of the protein within the splicing speckles in the nucleus, and identified a histidine stretch responsible for targeting the protein to this structure [5] in COS7 and Hela cells. Nevertheless, Nguyen and collaborators [52] showed that overexpressed DYRK1A in the brain of Tg(*Dyrk1a*)189N3Yah (Table 1) mice accumulates in the cytoplasm and the synapse. DYRK1A was found to bind to the WD40-repeat protein WDR68, this binding leading to their nuclear translocation and not requiring any kinase activity [53,54], whereas re-localisation of DYRK1A from the nucleus to the cytosol was found to occur when DYRK1A was bound to the brain-specific protein phytanoyl-CoA hydroxylase-associated protein 1 (PAHX-AP1) [55]. Moreover, different phosphorylated forms of DYRK1A were associated with different subcellular compartments, suggesting specific functions associated with specific phosphorylated forms and specific localisations [56].

Interestingly, analysis of de novo DYRK1A mutations found in MRD7 patients pointed at mutations outside of the kinase domain with no impact on the kinase activity, suggesting kinase-activity independent functions for DYRK1A [60]. In addition to this, a synergic Dyrk1a and Arip4 activating effect on androgen receptor- and glucocorticoid receptor-mediated transcriptional activation was observed independently of Dyrk1a kinase activity [61]. Hence, like other kinases, DYRK1A seems to have phosphorylation-independent functions such as scaffolding, allosteric regulation or protein-DNA interaction [62].

### 2.2. Targets and Interactors of DYRK1A

A consensus phosphorylation sequence has been determined for DYRK1A (R-P-x(1,3)-S/T-P) with DYRK1A having a preference for an arginine residue in the −2 or −3 position and for a proline at the + 1 position [63]. The specificity makes the binding different from the MAPK(PX(S/T)P). This sequence specificity has been validated in some DYRK1A target proteins such as TAU [64,65], amphiphysin [66] and caspase 9 [67]. However, phosphorylation sites that do not fit with this sequence have also been found in several DYRK1A substrates [30,68,69]. In a phosphoproteomic screen looking at differentially phosphorylated proteins in different brains structures of Tg(*Dyrk1a*)189N3Yah mice, we only found very few phosphopeptides displaying the DYRK1A consensus phosphorylation sequence, suggesting that Dyrk1a in Tg(*Dyrk1a*)189N3Yah differentially phosphorylates directly only very few targets, or that other DYRK1A phosphorylation sites might be overlooked [52].

Interactors and substrates of DYRK1A have helped to define the function of DYRK1A in different cellular processes and molecular cascades. To date, the BioGrid protein interaction network database listed 419 proteins presenting physical interaction with DYRK1A, most of them having been found through high-throughput affinity-capture mass spectrometry (https://thebiogrid.org/, accessed on 10 October 2021). In the next paragraphs, we will describe DYRK1A substrates and interactors involved in the regulation of different cellular processes throughout neurogenesis, brain function and neurodegeneration (see Figure 1 and Table 2).

#### 2.2.1. During Cell Proliferation and Neurogenesis

Consistent with the identification of DYRK family kinases as cell cycle regulators [70] and with the observation of microcephaly in humans, Drosophila and mice with MNB/DYRK1A haploinsufficiency [3,71,72,73,74] impaired neuronal proliferation have been reported in both loss and gain of function of DYRK1A as well as in DS individuals [75,76,77,78].

DYRK1A is a negative regulator of the G1-S phase transition. Overexpression of DYRK1A leads to an arrest of the cells in G1 phase, this lengthening of the G1 phase leading to cell cycle exit and promoting premature differentiation [76]. This cell cycle exit is triggered by upregulation of the cyclin-dependent kinase inhibitor p27KIP1 (CDKN1B) [79], which was achieved by its phosphorylation by DYRK1A at Ser10 leading to its stabilization [80]. The Drosophila homologs Mnb regulates the expression of Decapo, the Drosophila homolog of p27Kip1, by upregulating the expression of the proneural transcription factors asense and prospero in Drosophila newborn neuronal precursors, leading to their cell cycle exit and facilitating neuronal differentiation [81]. Another cyclin-dependent kinase inhibitor, p21CIP1 (CDKN1A), is also upregulated at the transcriptional level by DYRK1A overexpression, through phosphorylation of p53, leading to impaired G1/G0-S phase transition and attenuated proliferation of immortalized rat embryonic hippocampal progenitor H19-7 cells [30]. DYRK1A additionally controls nuclear levels of Cyclin D1 by phosphorylating this protein at Thr286, promoting its nuclear export and degradation [80,82]. DYRK1A phosphorylation of LIN52 at Ser28 is required for the assembly of the DREAM complex which coordinates gene expression during the cell cycle [83]. Formation of this complex is essential for the cell to enter quiescence or to undergo Ras-induced senescence, showing the importance of DYRK1A for the suppression of cell proliferation [83]. Levy and collaborators [84] further identified growth signaling pathways that are perturbed in mouse embryonic cortices missing *Dyrk1a* expression, including TrkB-BDNF and mTOR signaling.

DYRK1A can disrupt cell cycle progression by acting on microtubule dynamics in mitotic spindles of Xenopus tropicalis embryonic forebrain cells [144]. Furthermore, localization of Dyrk1a to ciliary axonemes in ciliated tissues suggests a role of DYRK1A in ciliogenesis [144]. DYRK1A might directly target β-tubulin phosphorylation in microtubule formation as regulation of microtubule dynamics occurs via the phosphorylation of this protein by MNB in Drosophila dendritic arborization neurons [113]. DYRK1A regulation of microtubule dynamics but also of other cytoskeletal proteins, such as actin, additionally plays a role in neuritogenesis and synaptogenesis. DYRK1A overexpression leads to reduction of dendritic branching and spines in the brain of DS mouse models and human patients [145,146,147,148], while underexpression of DYRK1A in leads to reduced neurite length and increased neurite branching [74,94,149,150]. DYRK1A overexpression promotes neurite outgrowth by forming a complex with RAS, B-RAF and MEK1, enhancing the mitogen-activated protein kinase cascade in PC12 cells [88], while knockdown of DYRK1A compromises neuritogenesis by decreased phosphorylation of the microtubule-associated protein 1B (MAP1B) at Ser1392 [94]. Contribution of DYRK1A to the regulation of actin dynamics has been observed in Yeast, with its *Schizosaccharomyces pombe* homolog interacting with the GTPase-activating protein RGA4 and contributing to the proper localization of RGA4 and the F-actin regulator CDC42 GTPase to cortical sites of the cell where F-actin structures are localized to generate cell polarity [95] and with its Mnb homolog identified in a kinome RNAi screen based on cell morphology as a modulator of actin-based protrusion [151]. Control of actin filament assembly by DYRK1A during neuronal dendrite and dendritic spines development occurs through multiple threonine phosphorylation of the N-WASP protein within its GTPase-binding domain, leading to the reduction of Arp2/3-mediated actin filament assembly [96].

#### 2.2.2. Synaptic Function

Many hints suggest that DYRK1A is not only implicated in neurogenesis but also plays a role in adult brain function. DYRK1A is expressed in mature brain neurons [20,21]. Deficits in learning and memory observed in DS mouse models as well as transgenic mice overexpressing *Dyrk1a* have been associated with increased long-term depression (LTD) at the expense of long-term potentiation (LTP) [152,153,154,155,156,157]. Moreover, decreased basal synaptic activity was measured in cortical primary cultures from Tg(*Dyrk1a*)189N3Yah mice [148] and increased in the amplitude of miniature excitatory postsynaptic currents (mEPSCs) was observed in pyramidal neurons of the prefrontal cortex of the same model [158]. Interestingly, we and our collaborators have observed, in the hippocampus of trisomic mouse models, a downregulation of some immediate-early genes (*Npas4*, *Arc*, *c-Fos* and *FosB*) encoding proteins involved in transduction signals that are induced in response to a wide variety of cellular stimuli and that are implicated in neuronal plasticity [159,160]. We also found a decrease in RHOA/ROCK signaling in the DS hippocampi that controls the building and maintenance of dendrites [160,161]. Maintenance of adult brain neural activity might occur via regulation by DYRK1A of the transcription factors NFAT [101] and CREB [44] that are both involved in synaptic function [162,163,164]. Moreover, DYRK1A phosphorylates the GRIN2A (NR2A) subunit of the N-methyl-D-aspartate receptor (NMDAR) at the post-synaptic membrane, modifying its activity and decreasing its internalization [115,116] and hence increasing its concentration at the post-synapse. Interestingly, DYRK1A have found to be associated with other post-synaptic proteins such as NR2B, PSD95, SYNGAP1 and CAMK2 [117]. DYRK1A might also regulate intracellular receptor trafficking via its interaction with SPROUTY, a protein involved in the control of receptor trafficking [51]. DYRK1A also phosphorylates or interacts with proteins known to play a crucial role in synaptic plasticity such as GSK3β [112], CAMKII [118] and RAS, RAF and MEK1 proteins from the MAPK/ERK signaling pathway [88]. DYRK1A was found to phosphorylate many proteins involved with membrane trafficking and vesicle endocytosis, a mechanism that is crucial for membrane recycling after neurotransmission release and hence for proper synaptic activity. DYRK1A regulates clathrin-mediated vesicle endocytosis via phosphorylation of the phosphatase synaptojanin I, the GTPase dynamin I, the scaffold protein amphyphysin I (AMPH) and the clathrin assembly proteins AP180, MAP1A, MAP2 and α- and β-adaptins [66,107,108,109,110,111,165]. Overexpression of DYRK1A in fibroblasts and neurons results in the inhibition of endocytosis due to defects in clathrin-mediated endocytosis [166]. Evidence for a role of DYRK1A at the pre-synapse comes from its localization and its tight coupling to the endocytic machinery [111,167]. DYRK1A was found to interact with and/or phosphorylate the presynaptic proteins MUNC18-1, a central regulator of SNARE-mediated exocytosis [114], the α-synuclein which chaperones the SNARE-complex assembly [134] and SYNAPSIN I involved in the maintenance of the reserve of the synaptic vesicle pool at the pre-synapse [52]. Interestingly, in an in-depth transcriptomic analysis of the hippocampus of six DS mouse models, Duchon and collaborators highlighted DYRK1A and SNARE as part of six central hubs connecting six major biological subnetworks deregulated in those models [160].

#### 2.2.3. Regulation of Expression (Methylation, Transcription, Translation)

Consistent with its presence in the nucleus, DYRK1A has a substantial number of target proteins that are transcription factors or proteins involved in gene expression. One mechanism of action of DYRK1A on those proteins is by altering their subcellular localization. DYRK1A negatively regulates NFAT by exporting it from the nucleus via its phosphorylation [100,101,102]. DYRK1A was shown to phosphorylate the GLI1 transcription factor involved in the Hedgehog signaling pathway at Ser408 in its NLS, promoting its nuclear import and its activation [89,90]. DYRK1A moves to the nucleus to interact with the intracellular domain of the Notch transmembrane receptor, promoting its phosphorylation in the Ankyrin domain and reducing its capacity to sustain transcription [91]. DYRK1A phosphorylates the FKHR transcription factor at Ser329, leading to a nuclear exit and a decreased gene transactivation [93]. DYRK1A also regulates gene transcription more directly by phosphorylating the C-terminal domain of RNA polymerase II (RNAPII) at Ser2 and Ser5 [105].

DYRK1A does not only mediate gene transcription via its action on transcription factors but also via the regulation of proteins involved in chromatin modification. DYRK1A phosphorylates the sirtuin SIRT1 at Thr522, promoting histone deacetylation [135]. DYRK1A was found to bind to the SWI/SNF chromatin remodeling complex known to interact with the neuron-restrictive silencer factor (REST/NRSF) and its overexpression led to perturbation of the expression of a cluster of gene located near the REST/NRSF binding sites and essential for neuronal differentiation [119]. DYRK1A interacts with the SNF2-like chromatin remodeling ATPase ARIP4 and synergistically activates androgen receptor-mediated and glucocorticoid receptor-mediated transcription [61]. DYRK1A also phosphorylates histone H3, antagonizing chromatin binding of the heterochromatin protein HP1 and leading to the transcriptional activation of genes involved in stress response including cytokines [120].

The presence of DYRK1A in the splicing factor compartment in nuclear speckles [5] has indicated a role of DYRK1A in RNA processing. DYRK1A was found to phosphorylate the splicing factor SF3b1/SAP155 at Thr434 both in vitro and in vivo [68]. DYRK1A also phosphorylates the alternative splicing factors ASF, SC35, SRp55 and 9G8, preventing inclusion of the alternatively spliced exon 10 of *Mapt/Tau* mRNA [121,122,123,124].

Recently, several studies have revealed an additional role of DYRK1A in DNA damage repair via its interaction with the ring finger protein RNF169, a negative regulator of double-strand breaks (DBS) repair, that leads to the recruitment of the 53BP1 scaffolding protein associated with non-homologous end joining (NHEJ)-promoting factor, at the DBS sites [125,126,127].

#### 2.2.4. Apoptosis, Neurodegeneration

In addition to being a neurodevelopmental disorder, DS is also characterized by accelerated aging. Signs of early aging are particularly prominent in the neurological system, with an extremely high prevalence of dementia and early onset of Alzheimer’s disease (AD) in DS [168]. Mouse models of DS recapitulate neurodegenerative DS phenotypes including the degeneration of basal forebrain cholinergic neurons and locus coeruleus noradrenergic neurons, neuro-inflammation and enhanced oxidative stress [169,170,171,172,173,174,175,176,177], but they do not present any Aβ deposits in their brain although studies reported increased of soluble and/or insoluble Aβ [175,178,179,180]. Although DYRK1A has been shown to have a protective effect against neuronal death, it also appears to be involved in neurodegenerative diseases including AD, Parkinson’s disease (PD) and Huntington’s disease (HD), and a link was found between DYRK1A and pro-inflammatory pathways that are activated in the brain of DS individuals and participate in the pathological mechanisms of neurodegenerative disorders. We will review here the interactions of DYRK1A with the different actors of those processes.

In 2008, Seifert and collaborators identified the cysteine aspartyl protease caspase 9 as a novel substrate of DYRK1A [132]. Caspase 9 is a critical component of the intrinsic or mitochondrial apoptotic pathway. The procaspase 9 is recruited to the apoptosome complex, where its dimerization by APAF-1 leads to its autocatalytic processing and activation [181]. Caspase 9 is activated by protein kinases of different transduction pathways initiated by extracellular signals or cellular stresses [182,183,184]. Active caspase 9 activates downstream effector caspases such as caspase 3 and caspase 7, leading to disassembly of the cell [185]. Seifert and collaborators showed that DYRK1A phosphorylates caspase 9 at its Thr125 in the cell nucleus and inhibits its auto-processing, setting a threshold for its activation through a basal inhibitory phosphorylation [132]. Moreover, they demonstrated that Caspase 9 inhibition by DYRK1A occurs in response to cell hyperosmotic stress [132]. Regulation of apoptotic cell death via this process was demonstrated in retinal progenitor cells [133]. A protective role of DYRK1A against apoptosis was also found in neuroprogenitor cells via phosphorylation of the huntingtin interacting protein 1 (HIP1), blocking HIP1-mediated cell death [142] and of the sirtuin SIRT1 on Thr522, triggering deacetylation of p53 and promoting cell survival [135].

Although neurodegeneration in AD was initially explained by the presence of the β-amyloid precursor protein (APP) gene on the Hsa21, increasing evidence points at a role of DYRK1A in neurodegenerative processes [99,186,187]. DYRK1A overexpression could be a primary risk factor contributing to the enhancement of β-amyloidosis as it was found to hyper-phosphorylate APP at Thr668 in vitro and in Tg(*Dyrk1a*)189N3Yah mice, increasing its cleavage by β- and γ-secretases and resulting in the formation of Aβ40 and Aβ42 peptides [130,131,188,189]. Moreover, DYRK1A was found to phosphorylate presenilin 1 (PS1), a key component of the γ-secretase complex, increasing γ-secretase activity and the generation of β-amyloid [137]. Finally, DYRK1A also phosphorylates the Aβ degradation enzymes neprilysin (NEP) and the insulin degrading enzyme IDE, decreasing their activity [138,140]. DYRK1A additionally contributes to neurodegeneration and AD pathology by a hyperphosphorylation of the microtubule-associated protein TAU, priming it to phosphorylation by GSK3 and leading to MAPT/TAU self-aggregation and the appearance of neurotoxic fibrillary tangles [64,128,129,190]. DYRK1A also contributes to tauopathies by the regulation of *Mapt/Tau* mRNA splicing through the phosphorylation of different splicing factors. Phosphorylation of the splicing factors ASF, SC35 and SRp55 by DYRK1A prevents the promotion of *Mapt/Tau* exon 10 inclusion by those factors leading to an overproduction of the 3R-TAU form at the expanse of the 4R-TAU, leading to of 3R/ 4R TAU ratio imbalance, exacerbating AD disease [121,123,124]. DYRK1A phosphorylates α-synuclein, a major component of the Lewi bodies (LB) that are found in several neurodegenerative disorders including Parkinson’s disease (PD), dementia with LB and AD, enhancing its aggregation within the cytoplasm of neurons and glial cells [69]. DYRK1A was also found to phosphorylate the PD-associated protein parkin at Ser131, inhibiting its ubiquitin E3 ligase activity and impairing the degradation of its substrates and leading to the induction of dopaminergic cell death [141]. Septin 4 (SEPT4), a GTPase previously found in neurofibrillary tangles of AD and in LB was also found to be a substrate of DYRK1A in neocortical neurons, expanding the mechanisms by which DYRK1A is implicated in neurodegenerative diseases [136]. DYRK1A is also associated with neuroinflammation which is implicated in neuronal damage and the onset of neurodegenerative diseases. Chronic glial cell activation results in the continuous release of proinflammatory cytokines, increasing neuronal death. DYRK1A was found to contribute to the development of the inflammatory microenvironment by removing the transcriptional repression on inflammatory cytokines by phosphorylating H3 and hence preventing the recruitment of the heterochromatin protein HP1 that normally inhibits the transcription of inflammatory interleukins [120]. Recently, Latour and collaborators [191] also found a link between DYRK1A and the proinflammatory NFkB signaling pathway, DYRK1A overexpression leading to the stabilization of IkBα protein levels by inhibition of calpain activity.

## 3. DYRK1A as a Target for Improving DS Cognition in Young Adults

DYRK1A has been the focus of many investigations where some groups demonstrated its involvement in cognitive dysfunction in DS models [160,192,193,194,195]. Several pieces of evidence showed that the decrease of DYRK1A activity to a normal level leads to the rescue of cognitive phenotypes in adult mice. Different strategies were used such as the expression of a short hairpin RNA against *Dyrk1a* transcripts introduced in the brain by adeno-associated viruses [196] or the treatment with drugs to reduce the kinase activity [197,198] or combining a full knock-out mouse with a DS model to go back to a normal gene dosage [195]. This rescue strategy emphasizes the major role of *Dyrk1a* overexpression in the appearance of the cognitive phenotypes observed in the partial DS mouse models. In this chapter, we decided to focus on drugs that target the protein DYRK1A. We selected those that have been tested in DS adult mouse models (Table 3) and for which relevant rescue results have been reported on cognition. Moreover, we will introduce some new DYRK1A inhibitors never tested on in vivo DS mouse models. 

### 3.1. Natural Product Derived DYRK1A Inhibitors and Their Derivatives

#### 3.1.1. Harmine, a Structural Analogue of β-Carbolin and an Indole Derivate

Harmine is an alkaloid β-carbolinic with a pyrido[3,4-b] indole ring structure isolated from *Banisteriopsis caapi*. It is a natural inhibitor of DYRK1A kinase activity [34]. Harmine is an ATP-competitive inhibitor of DYRK1A with a half maximal inhibitory concentration (IC50) of 0.08 µM [203]. Harmine was tested on two DS mouse models: (i) the Tg(*Dyrk1a*)189N3Yah model overexpressing *Dyrk1a* [204]; and (ii) the Ts(17^16^)65Dn model called Ts65Dn [205]. This model results from a reciprocal chromosomal translocation and is trisomic for 132 genes from the distal part of mouse chromosome (MMU)16 and for 60 genes from the centromeric part of MMU17 which are not orthologous to those of human chromosome 21 [206]. Harmine induced a decrease in homocysteine and in the Extracellular signal-Regulated Kinases (ERK) ½ level, due to a targeted effect on DYRK1A kinase activity [199] in the liver of Tg(*Dyrk1a*)189N3Yah mouse. Moreover, harmine prevents the maturation of neuronal progenitors in the Ts65Dn model [207]. At the molecular level, harmine produced a decrease of TAU phosphorylation in H4 neuroglioma cells overexpressing TAU [208]. TAU being a substrate of DYRK1A, this phosphorylation decrease reflects reduced DYRK1A kinase activity [128]. However, harmine is also known to target the monoamine oxidase-A (MAO-A), inhibiting the neuro-system circuit involved in behavior and mood control (serotonin, noradrenaline and dopamine axes) [209,210,211]. As such, harmine has neurotoxic side effects with the appearance of convulsions, involuntary movements and stiffness of the limbs. Accordingly, the use of harmine is not recommended on animal models or on human beings [212,213]. Consequently, derivatives of harmine have been synthetized to selectively inhibit DYRK1A without acting on MAO-A [211]. One of the optimized analogs, AnnH75, does not have effects on MOA-A activity, and still has the ability to decrease the phosphorylation of three known substrates of DYRK1A in HeLa cells: the splicing factor 3b1 (SF3B1) on Thr434, septin 4 (SEPT4) and TAU on Thr212. In addition, this inhibitor has been shown to be effective on the autophosphorylation of DYRK1A. The harmine derivative, AnnH75, did not show cytotoxicity in in vitro studies, but its efficiency and lack of neurotoxicity still need to be proven in in vivo studies.

#### 3.1.2. Epigallocatechin Gallate Is the Most Broadly Used DYRK1A Inhibitor

Epigallocatechin gallate (EGCG), the most abundant catechine of the green tea, is a DYRK1A non-competitive inhibitor with an IC50 of 0.33 µM [214] used in preclinical and in human clinical studies [197,198,200,215,216]. The supplementation of pure EGCG, or EGCG together with other catechines to mouse models overexpressing DYRK1A, such as Tg(*Dyrk1a*)189N3Yah, YACtg152F7 and Ts65Dn mice, and to human, improves cognition and functional connectivity [158,197,198,200]. However, the administration of EGCG (50 mg/day during seven weeks) to the Ts65Dn model did not correct the deficit observed in the Morris water maze test [215,217]. Moreover, the deficit observed in Ts65Dn mice in the Novel Object Recognition test (NOR) was not found in the study of Stringer et al. [215] so they could not conclude about the effect of EGCG on the recognition memory. These results suggest that some behavioral deficits observed in the Ts65Dn line are not strictly due to *Dyrk1a* overexpression by itself, showing that overexpression of *Dyrk1a* is not sufficient to induce cognitive deficits in mouse models of DS. In addition, EGCG can act on many other targets, such as vimentin (intermediate filament) receptors of laminin, or the receptor of endocannabinoids and can have many secondary effects in cell division, migration or in the regulation of the neurotransmission. Nonetheless, EGCG was used in a clinical study which lasted 12 months (realized from 2014 to 2015 in Barcelona, Spain). In this study, 84 people with DS, aged between 16 and 34 years old, received a dose of 9 mg/kg/day of EGCG, and showed significant improvement of their visual recognition memory, their ability to control inhibition and their adaptive behavior [200].

#### 3.1.3. Imidazolone DYRK1A Inhibitor

Another DYRK1A inhibitor is the Leucettine 41 (L41), a synthetic analogue of imidazolones derived from the alkaloids of the marine sponge *Leucettamine B*. Compared to Harmine and EGCG, L41 has a higher specificity for DYRK1A with an IC50 of 0.04 µM [218]. It acts as a competitive inhibitor for the fixation site of ATP. Administration of L41 during 19 days at a concentration of 20 mg/kg in adult mice overexpressing DYRK1A in various DS models (Tg(*Dyrk1a*)189N3Yah, Ts65Dn and Dp(16)1Yey [219]) leads to a correction of memory deficits evaluated by the NOR task and a normalization of DYRK1A kinase activity in the brain of those mice [52]. However, the L41 was not able to rescue the object location memory in the Tg(*Dyrk1a*)189N3Yah model, emphasizing that memories tested in location and object recognition tasks are not under the control of the same molecular mechanisms. Using the functional magnetic resonance imaging (MRI), a change in the functional connectivity of specific brain areas involved in cognitive processes such as the hippocampal formation, the thalamus, the pallidum and the cingulate cortex, was observed after 19 days of L41 treatment of Dp(16)1Yey mice. Moreover, a quantitative phosphoproteomic study, comparing Tg(*Dyrk1a*)189N3Yah and Ts65Dn mice treated with L41 and control mice, revealed new protein interactors of DYRK1A: the SYNAPSIN protein involved in the synaptic response, and other cytoskeletal components involved in axonal organization. Further co-immunoprecipitation and kinase activity assays show that DYRK1A phosphorylates directly SYNAPSIN1 on Ser551 [52]. In addition, L41 has shown a neuroprotective role against the cytotoxicity induced by over dosage of glutamate in extracts of mice hippocampal HT22 immortalized cells and against the neurodegeneration induced by the amyloid precursor protein (APP) in cultures of rat cortical cells [10]. A protective effect was again reported in a mouse model of Alzheimer’s disease after receiving an intra-cerebroventricular injection of amyloid-β_25-35_ peptides. L41 administration provides a protection mechanism to these mice against cognitive decline induced by the presence of the amyloid-β_25-35_ peptide in working memory and spatial memory [220].

### 3.2. Synthetic DYRK1A Inhibitors

The CX-4945, a benzo[c][2,6]naphthyridine-8-carboxylic acid derivative, is a synthetic ATP-competitive inhibitor of DYRK1A with an IC50 of 6.8 nM [201]. CX-4945 was tested on a mouse model overexpressing the human DYRK1A gene (Tg(*DYRK1A*)36Wjs). The inhibition of DYRK1A activity by CX-4945 in this model was validated by the normalization of Tau-pThr212, a phosphorylation known to be mediated by DYRK1A. No cognitive assays were performed in this study [128,201]. In addition, applying this inhibitor on HEK293 cultured cells revealed a decreased phosphorylation of some DYRK1A targets such as Tau-pThr212, Presenilin-1 (PS1) and APP-pThr668 [201]. Moreover, CX-4945 allows for rescuing the eye and wing defects observed in a *Drosophila larvae* model overexpressing DYRK1A [201]. However, because of the structural similarity of the CX-4945 with harmine, the use of this inhibitor on animal models or human should be monitored for any neurotoxicity.

A new family of azaindole derivate inhibitors, to which harmine belongs, was developed: F-DANDYs, with a 3,5-diaryl-7-azaindole structure [221]. Effectiveness of one of this fluorinated derivates (the 3-(4-fluorophenyl)-5-(3,4-dihydroxyphenyl)-1*H*-pyrrolo[2,3-*b*]pyridine) was reported in Ts65Dn mice after two weeks of intraperitoneal administration at a dose of 20 mg/kg. Indeed, a rescue of the spatial learning deficit observed in the Morris water maze task [202] was shown, revealing the therapeutic potential of this fluorinated member of the F-DANDY family. Moreover, DANDY’s ability to decrease DYRK1A kinase activity on Tau was proved in HEK293 cells [202].

### 3.3. Promising DYRK1A Inhibitors as Therapy for DS

#### 3.3.1. Benzothiazole Derivates DYRK1A Inhibitors

Over these last 15 years, several DYRK1A synthetic inhibitors were discovered. For most of these molecules, the in vivo effect on the cognition of DS mouse models has not been studied yet. For example, INDY is a synthetic inhibitor derivate from a dual specificity protein kinase (CLK) inhibitor called TG003. This derivate is three times more specific for DYRKs with an IC50 of 0.24 µM [9]. It was shown by Woods et al. (2016) that INDY can inhibit DYRK1A activity through the decrease of TAU-phosphorylation on Thr212 in COS7 cells [130]. Moreover, in another study by Arron et al. [101], INDY was shown to be able to restore the calcineurin-NFAT pathway initially inhibited by DYRK1A overexpression in HEK293 cells. The proINDY, a prodrug of INDY that is more lipophilic and has a better membrane absorption, showed the same results on TAU-pThr212 phosphorylation and on the calcineurin-NFAT pathway as INDY. Moreover, administration of proINDY to *Xenopus laevis* embryos allows for correcting the deformities in the eyes and head due to over activity of DYRK1A. Another derivative of INDY, called **BINDY** (benzofuro-fused INDY), was developed from the basic skeleton of the interacting area discovered from the crystal structure of the DYRK1A/INDY complex. With a change in the phenol group of INDY by a dibenzofuran, they produced the BINDY derivative which presents a higher specificity to DYRK1A with an IC50 of 25.1 nM [222].

#### 3.3.2. Compounds Acting on DYRK1A Kinase Activity Stability

FINDY (Floding Intermediate-selective inhibitor of DYRK1A) and CaNDY (CDC37 association inhibitor for DYRK1A) are two new synthetic inhibitors of DYRK1A acting on the stability and/or the maturation of DYRK1A. These inhibitors delete the auto-phosphorylation of DYRK1A by antagonizing the interaction between DYRK1A and the kinase specific co-chaperone CDC37. The consequence is a structural destabilization and the degradation of DYRK1A during the maturation process in living cells [39,223]. FINDY was also tested in vivo in a *Xenopus* embryo model that overexpresses DYRK1A leading to malformations of neural tissues [39]. These phenotypes were corrected after FINDY administration. CaNDY is another competitive inhibitor of DYRK1A activity with an IC50 of 7.9 nM [223]. This inhibitor has been discovered by screening kinase inhibitors in 293T cells using a fusion protein: CDC37 linked to a component of *Oplophorus* luciferase, the CDC37-nanoKAZ.

#### 3.3.3. Compounds Discovered from Functional Cellular Analysis

GNF7156 and GNF4877, which are aminopyrazine compounds, were identified by screening mice cells in order to identify molecules which induce the pancreatic β-cell proliferation [224]. This study shows that GNF7156 and GNF4877 inhibit DYRK1A with an IC50 of 100 and 6 nM, respectively [225].

Moreover, ALGERNON (Altered GEneRation of NeurONs) has been discovered in a screening of proliferation-enhancer activity compounds in neural stem cells (NSC). This drug aims to correct the alteration in neurogenesis found in DS models. This new inhibitor of DYRK1A has good pharmacokinetics with an IC50 of 77 nM [226].

## 4. Antenatal and Prenatal Treatment

DS is a neurodevelopmental disorder, characterized by brain malformations originating during development of the embryos. Thus, the possibility to reduce DYRK1A activity during early prenatal stages, to compensate the aversive effects of its overdosage (due to the presence of an additional copy), was tested by [195] using a genetic approach. The results obtained pave the way to carry early prenatal treatment with DYRK1A inhibitors. The goal is to obtain a stable compensation of the adverse effects due to DYRK1A overdosage for the rest of the life while treating very early during in utero development.

The first attempt to demonstrate the effect of prenatal intervention with a DYRK1A inhibitor was done using a DS mouse model which carries a YAC construct overexpressing *Dyrk1a*, and a small number of genes (*Pipg*, *Ttc3*, *Dscr9*, *Dscr3*). An EGCG diet (0.6 to 1.2 mg/day) was initiated at the beginning of pregnancy until adulthood [198]. The treatment induced a significant reduction of brain weight and thalamus-hypothalamus volume in transgenic treated animals compare to the non-treated Tg. Using a novel object paradigm, they showed that the EGCG treatment also rescues cognitive impairment. They also scored the working memory using a Y-maze paradigm, but the DS mice did not show any defect. At the molecular level, the mRNA of brain derived neurotropic factor, a member of the nerve growth factor family, and its plasma membrane receptor TRKB, were corrected to normal levels in the hippocampus of Tg mice. This protocol was the first proof of concept that modulation of DYRK1A, starting at early stages of development potentially, could correct phenotypes induced by DYRK1A overdosage. Nevertheless, certain questions remained and need to be addressed. Firstly, it was not possible to know if this correction occurred during embryogenesis or adulthood because the treatment was not continuous. Secondly, there was no analysis of the cellular, physiological or signaling pathways that were implicated, and, thirdly, they did not test if the treatment could have a similar effect on a more complex DS mouse model.

In order to go further, a new study was done with three different treatments and a dose of 50 mg EGCG/kg (estimate for a 25 g mouse eating 5 g per day). The first treatment consisted of administering the compound only during the gestation period until weaning. The second treatment was pursued until adulthood, and the third one was given only at adulthood stage [227]. EGCG was administered to Tg(*Dyrk1a*)189N3Yah mice and to the more complex Dp(16)1Yey DS mouse model. The authors analyzed the excitatory/inhibitory balance which has been found to be disrupted in the brain of individuals with DS and in DS mouse models. This disruption probably results in too much GABA-related inhibition. However, deficits in excitatory inputs and glutamatergic transmission could also contribute to this imbalance (for review, see [228,229]). Tg(*Dyrk1a*)189N3Yah and Dp(16)1Yey mice present several common phenotypes in excitatory/inhibitory balance, in particular an increased density of GAD67 neurons and an overexpression of hippocampal GAD67, an enzyme that synthesizes GABA for neuron activity. The hippocampal stratum radiatum of untreated Tg(*Dyrk1a*)189N3Yah mice showed an increased density of GAD67 neurons and an overexpression of hippocampal GAD67. Similar results were obtained with other GABAergic markers (GAD65, GABA transporter). The two prenatal treatments corrected both phenotypes. In addition, when the treatment was stopped at weaning, the correction was maintained in the animal up to the age of 69 days, whereas treating adult mice did not modulate those phenotypes. Conversely, the correction of excitatory markers was more pronounced than that of inhibitory markers when treatment was only done in adult. In Dp(16)1Yey mice, significant effects of prenatal treatments were observed on the ratio of GAD67/NEUN in the stratum radiatum or on GAD67/GAD65 expression in the hippocampus. Moreover, the prenatal correction of GABAergic pathways which is maintained from weaning to adulthood correlates with the rescue of novel object recognition. This suggests that DYRK1A hyperactivity at the early prenatal stage is the source of the perturbation of the excitatory/inhibitory balance found in adults as the inhibition of DYRK1A kinase activity during in utero development was able to restore an almost normal excitatory/inhibitory balance in trisomic mice. This restoration has a long-term beneficial effect until adulthood and was able to rescue the memory process in simple and complex DS mouse models.

The fact that ALGERNON, another DYRK1A ATP competitive inhibitor, could also rescue some DS phenotypes after in utero treatment, was encouraging [226]. In this study, the DS mouse model used was the Ts1Cje containing a triplicated region homologous to human chromosome 21, going from *Sod1* to *Mx1* [59], smaller than the ones in Ts65Dn or in Dp(16)1Yey. The authors demonstrated in vitro that DYRK1A phosphorylates CYCLIN D1, a protein required for progression through the G1 phase of the cell cycle, to induce its degradation. To support the hypothesis that DYRK1A hyperactivity in DS condition leads to an increased CYCLIN D1 degradation, they showed that a majority of DS-derived fibroblasts were in the G1 phase. They also showed that *Dyrk1A* knockdown with an siRNA in DS derived fibroblasts normalizes the population of cells in the G1 phase to euploid control levels. Moreover, in adult DS animals, ALGERNON increased the number of proliferating cells without affecting the rate of differentiation or number of differentiated cells compared with vehicle-treated. However, in a more interesting way, a shorter prenatal treatment, done only from embryonic day (E)10 to E15 with 10 mg/kg, was able to rescue the thinning of two cortical structures observed in untreated Ts1Cje embryos: the cortical plate, which includes migrating and differentiating neurons, and the intermediate zone, which contains migrating neurons and axon tracts. Prenatal ALGERNON administration in this mouse model presents also interesting long-term effects, in particular the rescue of working memory deficit in Y-maze, the hippocampus-dependent spatial memory in a Barnes maze and the associative memory in the contextual fear conditioning. Even if the decreased hippocampal neurogenesis in adult’s trisomic offspring persisted despite the treatment, this result indicates that only acting on DYRK1A overdosage during the early stages of development can correct cognitive phenotypes with long-term beneficial effects. These data also reinforce the idea that DYRK1A overdosage acts first during neurodevelopment and is central to DS neurologic deficits.

McElya et al. [230] decided to focus on another feature of DS, the craniofacial abnormalities. All humans with DS exhibit dysmorphic cranio-facial phenotypes. These abnormalities take place during prenatal morphogenesis and growth. Interestingly, the Ts65Dn DS mouse model exhibits similar craniofacial dysmorphology as observed in newborns with DS. In order to show the implication of DYRK1A in this craniofacial phenotype, a treatment with EGCG (200 mg/kg EGCG) was done at E7 and E8 by oral gavage of pregnant Ts65Dn mice. This short treatment corrected some of the neural crest cell phenotypes at E9.5 like the first pharyngeal arch volume and NCC number, with some craniofacial correction lasting until adulthood while the overall mandibular phenotype was not significantly improved. Curiously, a longer treatment (E0 to E9), with a lower dose (0.124 g/mL EGCG in water), did not have the same corrective effect; in particular, there was no significant improvement in the number of NCC in the pharyngeal arch 1.

Stagni et al. (2016) investigated the short- and long-time effects of postnatal inhibition of DYRK1A. The inhibitory treatment was administrated to Ts65Dn mice from postnatal day (P)3 to P15 during peak granule cell neurogenesis in the hippocampal dentate gyrus, with pure Sigma–Aldrich EGCG (25 mg/kg/day) [231]. The effect of the treatment was investigated immediately after the period of administration or at the age of 45 days, in order to detect possible long-term effects. At the cellular level and after the administration window, the authors reported that neonatal treatment with EGCG restored the number of neural precursor cells in two major brain neurogenic regions in trisomic model that are the sub-ventricular zone and sub-granular zone, without any effect on cell death. This led to an increase in total granule cell number reaching the disomic level. Surprisingly, the treatment had no effect in the euploid mice. They also found that the deficit in the number of proliferative cells in the Ts65Dn disappeared with the EGCG treatment. In the same way, EGCG restores the number of cells in the S-phase of the cell cycle. Because the neuronal circuit formation occurs in the early postnatal period, synapse development was also investigated. The levels of pre- and postsynaptic proteins in the dentate gyrus (DG), hippocampus and neocortex were decreased in Ts65Dn non-treated mice compared to the euploid non-treated mice and EGCG normalized these protein levels. Unfortunately, the treatment failed to induce long-term effects. When Ts65Dn mice treated with EGCG from P3–15 were examined at P45, there was no long-term improvement in neurogenesis. The proliferation rate of granule cell precursors, the number of granule neurons or the density of synaptic terminals in the hippocampus and neocortex were again impaired at 45 days of age. Memory performances were also evaluated at P45 to see if cognitive improvement of Ts65Dn mice could remain after a long period without treatment. Two hippocampal-dependent tasks were used, the working memory assessed in a Y-maze and the spatial reference memory assessed in the Morris water maze system. Both of them were not ameliorated. In addition, in treated Ts65Dn mice, a reduction in the number of BrdU-positive cells was observed compared to the Ts65Dn non-treated or euploid non-treated group of mice. There was also a reduction in granule cell numerical density in treated euploid mice. Even if no effect was noted in behavioral tests, these observations indicate potential aversive effects of the treatment.

These studies show that early prenatal (in utero) or antenatal treatment can lead to immediate correction of defects taking place during embryonic stages, whereas the sustainability of the effects does not always appear. Although many parameters differ between these studies, an obvious difference and surely of fundamental importance is the windows of treatment. Possibly because the treatment was given too late (after birth), it failed to produce long-term effect in the last study due to other alterations that are not compensated. In addition, we should be very careful with prenatal treatment. *Dyrk1a* is implicated in many primordial pathways and modifying its activity in a key period of development could lead to subtle impairments, undetectable in this simple animal model. The most prominent example is MRD7, a form of autism, due to a deficit in *Dyrk1a* gene dosage. Prenatal treatment should not mimic this deficit to avoid the apparition of deleterious phenotypes.

## 5. Conclusions

As discussed here, DYRK1A is a multipotent protein involved in several mechanisms during development and in adulthood (Figure). Many pieces of evidence have been collected over the last decades showing that DYRK1A overexpression is a major driver of cognitive phenotypes in DS and that DYRK1A controls the development of the brain and the function of the adult brain, notably at the glutamatergic synapses [52,117]. Several drugs have been developed with some success to reduce the hyperactivity of the kinase in DS and rescue the cognitive deficits in adults or after treatment in utero [230,231]. Changes in other domains have been investigated. Facial dysmorphology was improved in DS mouse models and also observed in young children treated with GTE-EGCG [230,232]. Nevertheless, Stringer et al. [215] stressed the negative effect on long bones development in treated mice. Recent studies pinpointed the consequences of treatment with EGCG in body composition and lipid metabolism in humans [233]. Interestingly, inhibitors of DYRK1A have been found to impact adipogenesis in a cellular differentiation model [222] and to promote the proliferation of pancreatic β-cells [225]. Most of the work done so far has considered that DYRK1A overdosage is constant during development and in adulthood. Nevertheless, it is clear from recent advances done with single cell analysis that the expression, and thus the overdosage of *Dyrk1a*, is not constant during mouse development. Thus, one will have to consider revisiting and refining the window of treatment for the embryonic development although no major defect was observed in mice when treated from conception to birth.

In addition, we should keep in our mind that reducing the DS etiology to the sole overdosage of DYRK1A is certainly not reasonable. DYRK1A is definitely not the only driver for DS phenotype. DS is a multigenic diseases with many overdosage genes, such as *Kcnj6 (Potassium Inwardly Rectifying Channel Subfamily J Member 6), App (Amyloid-β precursor protein)*, *(Cystathionine β synthase)* and others, contributing to cognitive phenotypes [160,234,235]. Thus, controlling the DYRK1A kinase overactivity in DS should remind readers as a real proof-of-concept that improving the life of people with DS can be reached. Additional progress needs to be achieved first to fine-tune the correction of DYRK1A overdosage and to disentangle the complex genetic interactions that are found in DS models, involving *DYRK1A* and other Hsa21 genes.

## Figures and Tables

**Figure 1 genes-12-01833-f001:**
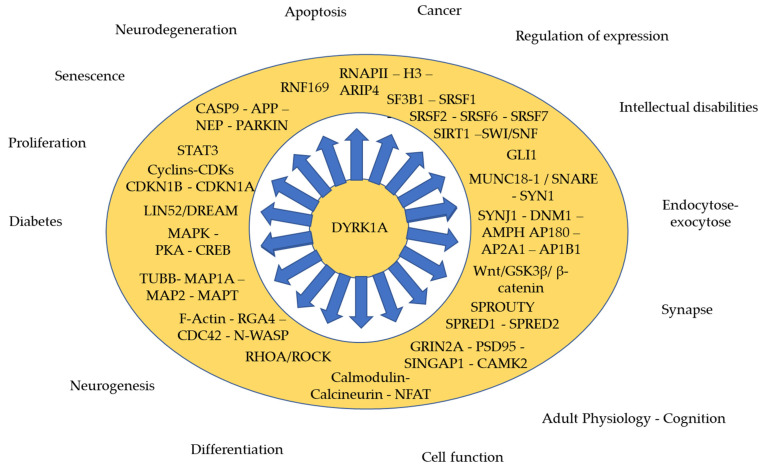
DYRK1A involved in various cellular processes during development and throughout the adult lifetime.

**Table 1 genes-12-01833-t001:** Transgenic line overexpressing *Dyrk1a* in the mouse.

Allele Symbol	Category—Genetic Background	Initial Reference	Repository
Tg(CEPHY152F7)12Hgc	Transgenic YAC Human- FVB or C57BL/6J including TTC3 (tetratricopeptide repeat domain 3), DSCR3 (Down syndrome critical region gene 3), DYRK1A (dual-specificity tyrosine-(Y)-phosphorylation regulated kinase 1A) and the potassium inwardly-rectifying channel, subfamily J, member 6 gene, (KCNJ6)	[57]	INFRAFRONTIER Stock EM:01303
Tg(MT1A-*Dyrk1a*)9Xest	Transgenic-Heterogenous promoter rat cDNA for *Dyrk1a* (2 copies)	[58]	Unknown
Tg(MT1A-*Dyrk1a*)33Xest	Transgenic-Heterogenous promoter rat cDNA for *Dyrk1a* (5 copies)	[58]	Unknown
Tg(*DYRK1A*)36Wjs	Transgenic BAC Human locus for *DYRK1A*—C57BL/6J coisogenic	[58]	JAX
Tg(*Dyrk1a*)189N3Yah	Transgenic BAC Mouse lous for *Dyrk1a*- C57BL/6J coisogenic	[59]	INFRAFRONTIER stock EM:02119

**Table 2 genes-12-01833-t002:** Main DYRK1A interactors, partners and targets (*only described in yeast).

Symbol	Name	Cell Function	Ref.
Neurogenesis			
CCND1	Cyclin D1	Cell cycle and division (G1 to S phase transition)	[80,82,85]
CCNL2	Cyclin L2	transcription regulation	[86]
CDKN1A	Cyclin Dependent Kinase Inhibitor 1A	Cell cycle and division (G1 to S phase transition)	[30]
CDKN1B	Cyclin Dependent Kinase Inhibitor 1B	Cell cycle and division (G1 to S phase transition)	[79,80]
LIN52	Protein lin-52 homolog	Cell cycle, transcription	[83]
SNR1	Integrase interactor	Cell cycle, transcription regulation	[87]
FGF2	Fibroblast Growth Factor 2	Mitogenic and angiogenic activity	[44]
RAS	GTPase Ras	Differentiation, transcription regulation	[88]
BRAF	B-Raf Proto-Oncogene, Serine/Threonine Kinase	Differentiation, transcription regulation	[88]
MEK1	Dual specificity mitogen-activated protein kinase 1	Differentiation, transcription regulation	[88]
GLI1	Glioma-associated oncogene 1	Differentiation, transcription regulation	[89,90]
NOTCH	Notch receptor	Differentiation, transcription regulation	[91]
GSK3b	Glycogen synthase kinase 3 β	Differentiation, neurogenesis	[92]
FKHR	Forkhead Box O1	Apoptosis, differentiation, transcription regulation	[93]
MAP1B	Microtubule Associated Protein 1B	Microtubule Dynamics, neuritogenesis, axon extension, intracellular transport	[94]
RGA4 *	Rho GTPase-activating protein 4	Actin filaments assembly	[95]
CDC42 *	Cell Division Cycle 42	Actin filaments assembly	[95]
N-WASP	Neural Wiskott-Aldrich syndrome protein	Actin filaments assembly, synaptogenesis, cell cycle, cell division, mitosis, transcription	[96]
GRB2	Growth Factor Receptor Bound Protein 2	Cell differentiation	[97,98]
Adult brain function/ Synaptic activity			
YWHAZ	14-3-3 kinase-binding protein	Signal transduction, spine maturation	[40,99]
NFATC	Nuclear factor of activated T cells	Synaptic plasticity, transcription regulation	[100,101,102]
RCAN1	Regulator of calcineurin 1	calcineurin-NFAT signaling cascade	[103,104]
CREB	CAMP-Responsive Element-Binding Protein	Synaptic plasticity, differentiation, transcription regulation	[44]
RNAPII	RNA polymerase II	Transcription	[105]
DNM1	Dynamin 1	Clathrin-mediated vesicle endocytosis	[106,107,108]
AMPH	Amphiphysin 1	Clathrin-mediated vesicle endocytosis	[66,107]
SH3GL2	SH3-domain GRB2-like 2/ Endophilin 1	Clathrin-mediated vesicle endocytosis	[109]
SYNJ1	Synaptojanin 1	Clathrin-mediated vesicle endocytosis	[107,110,111]
SNAP91	Clathrin Coat Assembly Protein AP180	Clathrin-mediated vesicle endocytosis	[107]
GSK3β	Glycogensynthasekinase3 β	Synaptic plasticity	[112]
MAP1A	Microtubule Associated Protein 1A	Clathrin-mediated vesicle endocytosis	[107]
MAP1B	Microtubule Associated Protein 1B	Axon extension, intracellular transport	[94]
MAP2	Microtubule Associated Protein 2	Clathrin-mediated vesicle endocytosis	[107]
TUBB	Tubulin β Class I/β-tubulin	Microtubule	[113]
AP2A1	Adaptor Related Protein Complex 2 Subunit α/1α-adaptin	Clathrin-mediated vesicle endocytosis	[107]
AP1B1	Adaptor Related Protein Complex 2 Subunit β 1/β-adaptin	Clathrin-mediated vesicle endocytosis	[107]
STXBP1	Syntaxin Binding Protein 1	SNARE-mediated endocytosis	[114]
SYN1	Synapsin I	Neurotransmitter release	[52]
GRIN2A	Glutamate receptor, ionotropic, NMDA2A	Receptor, ion transport	[115,116]
GRIN2B	Glutamate receptor, ionotropic, NMDA2B	Receptor, ion transport	[117]
SPRY2	Sprouty 2	Receptor trafficking	[51]
SPRED1	Sprouty Related EVH1 Domain Containing 1	RTK/Ras/ERK pathway	[43]
SPRED2	Sprouty Related EVH1 Domain Containing 2	RTK/Ras/ERK pathway	[43]
CAMK2	Calcium/Calmodulin Dependent Protein Kinase II	Synaptic plasticity	[118]
DLG4	discs large MAGUK scaffold protein 4 /Post-Synaptic Density Protein 95	Synaptic plasticity	[117]
SYNGAP1	Synaptic Ras GTPase Activating Protein 1	Synaptic plasticity	[117]
SWI/SNF	SWI/SNF-Related Matrix-Associated Actin-Dependent Regulator Of Chromatin Subfamily	Chromatin remodeling	[119]
ARIP4	Androgen Receptor-Interacting Protein 4	Chromatin remodeling, ATP, DNA and Nucleotide-binding	[61]
H3	Clustered Histone 14	Chromatin, chromosomal stability	[120]
SF3B1	Splicing factor 3b, subunit 1	mRNA processing, mRNA splicing	[68]
SRSF1	Serine/Arginine Rich Splicing Factor 1	mRNA processing, splicing and transport	[121,122,123,124]
SRSF2	Serine/arginine-rich splicing factor 2	mRNA processing, splicing and transport	[121,122,123,124]
SRSF6	Serine/arginine-rich splicing factor 6	mRNA processing, splicing and transport	[121,122,123,124]
SRSF7	Serine/arginine-rich splicing factor 6	mRNA processing, splicing and transport	[121,122,123,124]
RNF169	Ring Finger Protein 169	Repair following DNA damage	[125,126,127]
Neurodegeneration/Cell death			
MAPT	Microtubule-associated protein	Microtubule dynamics	[64,128,129]
APP	Amyloid precursor protein	Apoptosis, cell adhesion, endocytosis	[130,131]
CASP9	Caspase 9	Apoptosis	[132,133]
SNCA	α-synuclein	SNARE-mediated endocytosis	[134]
LATS2	Large Tumor Suppressor Kinase 2	Cell proliferation, apoptosis	[45]
SIRT1	Sirtuin 1	Histone deacetylation/ cell survival	[135]
SEPT4	Septin 4	Nucleotide-binding,	[136]
PSEN1	Presenilin 1	Neuronal degeneration, apoptosis, cell adhesion, Notch signaling	[137]
NEP1	Neprilysin	Aβ-degrading enzyme	[138,139,140]
IDE	Insulin Degrading Enzyme	Intercellular peptide signaling	[138,139,140]
PARK2	Parkin	Autophagy, transcription regulation	[141]
HIP1	Huntingtin interacting protein 1	Apoptosis, differentiation, endocytosis, transcription regulation	[142]
HAP1	Huntingtin Associated Protein 1	Vesicular trafficking	[143]
DCAF7	DDB1 and CUL4 associated factor 7	Ubl conjugation pathway	[53,54,143]
CAPN1	Calpain 1	Protease activity	[47]
STAT3A	Signal Transducer And Activator Of Transcription	Activator neuroinflammation	[48]
PHYHIP	Phytanoyl-CoA α-hydroxylase associated protein 1	DYRK1A interacting protein	[55]

**Table 3 genes-12-01833-t003:** Targeting DYRK1A activity using specific inhibitors in vivo with DS models.

DYRK1A Inhibitors	Animal Models Tested	Effects on Animal Models	Clinical Trial	Ref.
Harmine	Tg(*Dyrk1a*)189N3Yah	Decrease in homocysteine and ERK ½ liver	No information	[199]
	2 months age, 10 mg/kg, intra-periotoneal (i. p.) daily injection			
EGCG	YACtg152F7	Restoration of brain weight and morphology and object recognition memory		[198]
	From gestation to adulthood, orally administratation at 0.6–1 mg/day or 1.2 mg/day			
	Tg(*Dyrk1a*)189N3Yah	Normalization of the long term potentiation and spine density in the prefrontal cortex.		[158]
	3–4 months age, oral administration at 200 mg/kg/day during 4–6 weeks.			
	Tg(*Dyrk1a*)189N3Yah	Improvement of novel object recognition memory and spatial learning and memory.	2014, 12 months, Barcelona, Spain, Phase 2 trial. done on 84 people aged between 16 and 34 years old which received a dose of 9 mg/kg/day of EGCG. Improvement of visual recognition memory, ability to control inhibition and their adaptative behavior.	[197]
	3 months age, administration in drinking water during 1 month			

	Ts(17^16^)65Dn	Spatial learning and memory recovery.		
	3 months age, administration in drinking water during 1 month			
			2016, Barcelona, Spain, Phase 2 trial.	[200]
			Realize on 76 children (6 and 12 years old) which recerved a daily dose of 10 mg/kg/day of EGCG	
Leucettine 41	Tg(*Dyrk1a*)189N3Yah	Improvement of novel object recognition memory	Not done	[52]
	3 months age, i. p., administration during 19 days at 20 mg/kg			
	Ts(17^16^)65Dn	Improvement of novel object recognition memory		
	3 months age, i. p., administration during 19 days at 20 mg/kg			
	Dp(16)1Yey	Improvement of novel object recognition memory		
	3 months age, i. p., administration during 19 days at 20 mg/kg	Change in functional connectivity in brain area		
CX-4945	Tg(*DYRK1A*)36Wjs	Normalization of TAU-pThr212	Not tested	[201]
	8 to 10 weeks old, Acute oral administration of 75 mg/kg			
	Drosophilia larvae	Rescue eyes and wings deformities		
	Administration of 100 nM from embryo stage until adulthood stage			
F-DANDY	Ts(17^16^)65Dn	Rescue of spatial learning deficit	Not tested	[202]
	Administration by i. p., at 20 mg/kg/day			
INDY	*Xenopus laevis*	Eyes deformities correction	Not tested	[9]
	Embryos treated with 2.5 µM			
FINDY	*Xenopus*	Restoration of neural tissu malformation	Not tested	[39]
	Embryos treated with 2.5 µM			

## Data Availability

All the data used for this review are available as indicated in the text in online resource.

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
