# Peer review of "Dyrk1a from Gene Function in Development and Physiology to Dosage Correction across Life Span in Down Syndrome"

_genes, 2021, doi:10.3390/genes12111833_

Round 1
Reviewer 1 Report
This review in Genes is focused on Dyrk1a and its role in Down syndrome. This is a timely topic that is important for future development of drug targets for this vulnerable population. Dyrk1a is involved not only in neural development but also in aging processes and maintenance of function. The authors propose the idea that developmental intellectual disability could possibly be targeted by reducing Dyrk1a overexpression by the use of for example specific kinase inhibitors. This is a timely review of Dyrk1a data and only a few minor suggestions in terms of revisions. These are listed below:
- There are no tables or illustrations – the addition of summaries in the form of tables and a schematic overview figure of proposed Dyrk1a actions would enhance the readability of the review.
- Line 41: Many parents and other family members don’t consider Down syndrome a “disease” -instead they celebrate the special contribution that those with DS gives to society and their families. One should be careful about calling it an “incurable disease” and instead consider focusing on limiting or halting detriments associated with Down syndrome without trying to eradicate those with Down syndrome.
- Lines 52-54: Consider including a schematic drawing here for pathways that are regulated by Dyrk1a – this could make this section more clear.
- Line 72 Spell out Indy the first time it appears.
- Lines 77-90: Consider a table here to summarize publications on developmental Dyrk1a effects.
- g. line 200: There is mention of several different mouse models for Dyrk1a dysregulation – both transgenic and non-transgenic mice. A table summarizing the current models and publications focused on the different models would clarify for the reader which models exist and how these can be purchased.
- Line 288-300: A summary table showing Dyrk1a effects with citations would help summarize this portion of the paper.
- The same is true for lines 325-26: “part of six central hubs connecting six major biological subnetworks deregulated in those models”. It would be nice to have a schematic or table summarizing these six major subnetworks to illustrate their importance here.
- Line 365: “Signs of early aging are particularly prominent in the neurological system, with an extremely high prevalence of dementia and early onset of Alzheimer’s disease”. Do the authors mean “in Down syndrome”? If so, it should be added to the sentence for clarity.
- Section 2.2.4 is a bit tedious and contains a whole lot of lists in terms of what role Dyrk1a plays in neurodegeneration. This section could be a bit more fluent and contain a summary sentence or two in the end of it to summarize the Dyrk1a role for degeneration. Alternatively, a table showing the many different Dyrk1a activities and protein accumulation that are affected by Dyrk1a.
- Section 2.1 could also use a summary Table showing the different derivatives and if they are in clinical trials (include citations and clinical trial number or website).
- Section 5 (Conclusions) could also contain a summary schematic drawing or table summarizing the many different actions of Dyrk1a and a conclusion why targeting this protein might affect both developmental and age-related degeneration occurring in those with Down syndrome.
Author Response
Reviewer 1 :
This review in Genes is focused on Dyrk1a and its role in Down syndrome. This is a timely topic that is important for future development of drug targets for this vulnerable population. Dyrk1a is involved not only in neural development but also in aging processes and maintenance of function. The authors propose the idea that developmental intellectual disability could possibly be targeted by reducing Dyrk1a overexpression by the use of for example specific kinase inhibitors. This is a timely review of Dyrk1a data and only a few minor suggestions in terms of revisions. These are listed below:
- There are no tables or illustrations – the addition of summaries in the form of tables and a schematic overview figure of proposed Dyrk1a actions would enhance the readability of the review.
YH> We agree and we have inserted one figure and three tables (please see the attachement
- Line 41: Many parents and other family members don’t consider Down syndrome a “disease” -instead they celebrate the special contribution that those with DS gives to society and their families. One should be careful about calling it an “incurable disease” and instead consider focusing on limiting or halting detriments associated with Down syndrome without trying to eradicate those with Down syndrome.
YH> We agree with the comment we do not deny that people with DS contributes and gives to society and their families. Nevertheless, we should consider the low autonomy, the intellectual disabilities, the different comorbidities attached to DS that affect people with DS as disease for which at the moment we have no cure to leverage the burden. An incurable disease does not mean an eradication of those with Down syndrome. I could propose to change the term “incurable disease” to “ limiting and detrimental diseases”.
- Lines 52-54: Consider including a schematic drawing here for pathways that are regulated by Dyrk1a – this could make this section more clear.
YH> in this lane 52-54 we summarize the roles of the CMGC kinase group.( “roles in cell cycle regulation, intracellular signal transduction, cell growth and communication.”). We don’t really see the need of a scheme here that will just reproduce the text. To help the reader we introduced a new reference Varjosalo et al 2013 that describe the Human CMGC kinase group.
- Line 72 Spell out Indy the first time it appears.
YH> Done
- Lines 77-90: Consider a table here to summarize publications on developmental Dyrk1a effects.
YH> In the line 77-90 we only summarize the expression of DYRK1A in neuronal progenitors and in the brain during late development of adult. We do not discuss the developmental effect of Dyrk1a.
- g. line 200: There is mention of several different mouse models for Dyrk1a dysregulation – both transgenic and non-transgenic mice. A table summarizing the current models and publications focused on the different models would clarify for the reader which models exist and how these can be purchased.
YH> We have inserted the full list of transgenic models overexpressing Dyrk1a with place where some can be found
- Line 288-300: A summary table showing Dyrk1a effects with citations would help summarize this portion of the paper.
YH> We agree and we have proposed one figure showing the main interactors of DYRK1A and we have inserted one table
- The same is true for lines 325-26: “part of six central hubs connecting six major biological subnetworks deregulated in those models”. It would be nice to have a schematic or table summarizing these six major subnetworks to illustrate their importance here.
YH> major subnetworks RHOA, SNARE AND GSK3b involving Dyrk1a are indicated in the figure
- Line 365: “Signs of early aging are particularly prominent in the neurological system, with an extremely high prevalence of dementia and early onset of Alzheimer’s disease”. Do the authors mean “in Down syndrome”? If so, it should be added to the sentence for clarity.
YH>We agree and we changed the sentence to “Signs of early aging are particularly prominent in the neurological system, with an extremely high prevalence of dementia and early onset of Alzheimer’s disease (AD) in DS”
- Section 2.2.4 is a bit tedious and contains a whole lot of lists in terms of what role Dyrk1a plays in neurodegeneration. This section could be a bit more fluent and contain a summary sentence or two in the end of it to summarize the Dyrk1a role for degeneration. Alternatively, a table showing the many different Dyrk1a activities and protein accumulation that are affected by Dyrk1a.
YH> We included the targets and interactor of DYRK1A in the figure and the table. Most of the changes described are phosphorylation and not always associated with protein accumulation.
- Section 2.1 could also use a summary Table showing the different derivatives and if they are in clinical trials (include citations and clinical trial number or website).
YH> table 3 shows now the information requested
- Section 5 (Conclusions) could also contain a summary schematic drawing or table summarizing the many different actions of Dyrk1a and a conclusion why targeting this protein might affect both developmental and age-related degeneration occurring in those with Down syndrome.
YH> We think that the new figure recapitulates the information

Reviewer 2 Report
This is a comprehensive review concerning functional studies of DYRK1A and therapeutic approaches targeting its gene product. Overdosage of DYRK1A plays a major role in the pathogenesis of Down syndrome, and haploinsufficiency of this gene causes a separate form of intellectual disability.
Comments/criticisms:
Nomenclature: The nomenclature of the gene and protein, i.e. the use of italics and the differentiation between the gene and protein in humans and in mice, is not consistent.
In the introduction, I suggest including a sentence that DYRK1A is part of the Down syndrome critical region (DSCR).
Line 33: The “D” in MRD7 stands for dominant, not for disease (Mental retardation, autosomal dominant 7).
Line 108: What does “… P44, a shortest, …” mean?
The manuscript needs some language editing, e.g. line 78 “mains proteins” (sic); line 90 “extend” (sic).
Author Response
Reviewer 2:
This is a comprehensive review concerning functional studies of DYRK1A and therapeutic approaches targeting its gene product. Overdosage of DYRK1A plays a major role in the pathogenesis of Down syndrome, and haploinsufficiency of this gene causes a separate form of intellectual disability.
Comments/criticisms:
Nomenclature: The nomenclature of the gene and protein, i.e. the use of italics and the differentiation between the gene and protein in humans and in mice, is not consistent.
YH> We revised the text and hope to match now the current nomenclature. We included one new figure and three new tables (please see the attachement)
In the introduction, I suggest including a sentence that DYRK1A is part of the Down syndrome critical regio (DSCR).
YH> the concept of Down syndrome critical region (DSCR) has been unvalidated by many experiments and in particular by Olson et al 2004. We do not want to confuse the reader with an old concept
Line 33: The “D” in MRD7 stands for dominant, not for disease (Mental retardation, autosomal dominant 7).
YH> Corrected. Thanks
Line 108: What does “… P44, a shortest, …” mean?
YH > we corrected for “shortest form”
The manuscript needs some language editing, e.g. line 78 “mains proteins” (sic); line 90 “extend” (sic).
YH> We corrected many typos but we did not have enough time to send the manuscript for review by a proof-reading service. If the editors agree to give at least a week we can do it.
